# Assessing Reproductive Performance to Establish Benchmarks for Small-Holder Beef Cattle Herds in South Africa

**DOI:** 10.3390/ani12213003

**Published:** 2022-11-01

**Authors:** Marble Nkadimeng, Este Van Marle-Köster, Nkhanedzeni Baldwin Nengovhela, Fhulufhelo Vincent Ramukhithi, Masindi Lotus Mphaphathi, Johannes Matthias Rust, Mahlako Linah Makgahlela

**Affiliations:** 1Department of Animal and Wildlife Sciences, University of Pretoria, Hatfield, Pretoria 0002, South Africa; 2Agricultural Research Council, Department of Germplasm Conservation and Reproductive Biotechnologies, Private Bag X2, Irene, Tshwane 0062, South Africa; 3Department of Agriculture, Land Reform and Rural Development, Delpen Building, Corner Annie Botha and Union Street, Riviera, Pretoria 0001, South Africa; 4Department of Agriculture and Animal Health, University of South Africa, Florida 1710, South Africa; 5Döhne Agricultural Development Institute, Stutterheim 4930, South Africa; 6Department of Animal, Wildlife and Grassland Sciences, University of the Free State, Bloemfontein 9301, South Africa

**Keywords:** cow fertility, management factors, performance benchmarks, pregnancy rate

## Abstract

**Simple Summary:**

In South African beef cattle smallholder farms, there has been no recommended target benchmark that provide a baseline for improving the reported low herd reproductive performances. A multi-stage sampling approach was performed to examine reproductive performance as defined by pregnancy rate, fetal and calf losses, calving interval and days open to benchmark smallholder herd reproduction. It was found that smallholder farms recorded on average, 50% pregnancy rate and 12% fetal and calf losses, with days open and calving interval achieved at 334 and 608 days, respectively. Targeted benchmarks for performance derived from this study were 54%, 1.4%, 152 and 425 days, respectively for pregnancy rate, fetal and calf losses, days open and calving interval for smallholder farms in South Africa. The study showed that herd management practices including non-culling of old and non-productive cows, no knowledge of body condition score prior to breeding, no record keeping, continuous breeding season and low bull to cow ratio are associated with recorded reproductive performance norms in smallholder farms. The study found that smallholders have the potential to improve their performance levels if management knowledge is provided through advisory and extension services.

**Abstract:**

Smallholder beef cattle farms in South Africa have had low reproductive performance, which has been associated with management practices. Considering current farm management practices, a multi-stage selection study was conducted to assess reproductive performance as defined by pregnancy rate, fetal and calf losses, calving interval and days open to benchmark reproductive performance. Data were collected twice, in autumn (March–May) for pregnancy diagnosis and in spring (September–November) for monitoring of confirmed pregnancies. Overall, 3694 cow records from 40 smallholder herds were collected during 2018 and 2019 breeding seasons from five provinces. The preferred 25th quartile described target performance and GLIMMIX procedure determined associations between management practices and performance. Smallholder farms on average recorded 50% pregnancy rate and 12% fetal and calf losses with 304 and 608 days open and calving interval, respectively. The derived target benchmarks for pregnancy rate, fetal and calf losses, days open and calving intervals in smallholder farms were 54%, 1.4%, 152 and 425 days, respectively. Reproductive performance was associated with no knowledge of body condition scoring before breeding, culling of old and non-productive cows, record keeping and low bull to cow ratio (*p* < 0.05). The performance benchmarks implied that industry averages may be improved if sustainable management services are provided through extension and advisory services.

## 1. Introduction

The potential of smallholder farmers on eradicating poverty and improving food security in rural communities of most African countries including South Africa (SA) has been well-recognized [1,2]. The smallholder sector is a driving force of farming in developing countries. In sub-Saharan Africa and Asia, 80% of the food supply is produced by smallholder farmers [3]. Worldwide, the sector supplies 60% of meat and 75% of dairy produce [4,5]. Thus, the improvement of this sector towards a sustainable farming system can respond to multiple Sustainable Developmental Goals [6].

In livestock production, a sustainable farming system is characterized by improved herd productivity and profitability. Reproductive performance is one of the factors influencing farm productivity because successful pregnancy and parturition rates are drivers of farm profit [7,8]. In South African smallholder farms, reproductive performance of beef cattle under extensive systems has been reported as low for over a decade with average calving rates of ≤48% [9,10,11,12,13]. This figure is lower than the established industry standard of 65% calving rate in commercial herds and the department of agricultures’ recommended national average of 85% for beef cattle in SA [14,15]. To date, beef cattle farming in SA smallholder farms reports no measures of herd selection for reproductive performance indicators. Moreover, lack of understanding of basic herd management principles and uncontrolled breeding systems are a norm to majority of the farms [7,16,17,18].

Calving rate has been utilized as the single and most prominent indicator to define reproductive performances in SA smallholder herds [12,13,19,20]. However, as an effective measure of production, calving rate may have limitations in detecting underlying reproduction components. For example, assessing early warnings of reproductive diseases such as trichomoniasis and brucellosis, as well as reproduction challenges such as infertility in males and females [21]. A reflection of good herd reproduction is an indication of successful cow conception to produce viable offspring within an acceptable timeframe [22]. Therefore, there is a need to define a set of indicators, which to an extent may provide a comprehensive summary assessment defining herd reproductive performances from conception to calving. This is to provide a greater understanding of herd reproductive performance and reveal areas that require attention [23,24]. Assessment of indicators such as pregnancy rate, days open, calving interval and pregnancy losses collectively can provide detailed performance levels of fertility in the herds [18,25]. Selection to improve these performance indicators in smallholder farmers has been predicted to promote participation in designing efficient on-farm community-based breeding systems [26]. However, knowledge of herd management practices is required in understanding performance benchmarks for these indicators in smallholder farms [27].

Previous research reporting on reproductive performance in SA smallholder farms relied on farmer questionnaires and surveys. These studies are dependent on farmers’ memories of their herd performances as recording has not been adequately prioritized in smallholders [11,16,28,29]. Added to these, assessments of reproductive performance are focused on single areas and this is prohibiting a holistic view of performance at national, herd and animal level [29]. The current research acknowledges these gaps and attempts to study current breeding practices by evaluating multiple performance indicators from on-farm animal records at an extended geographic area to broaden information within the SA smallholder farms. The research aims at assessing reproductive performance as defined by pregnancy rate (PR), fetal and calf losses (FC), calving interval (CI) and days open (DO) on beef cattle farms to set benchmarks for herd reproductive performance. Furthermore, the study aims to assess whether management practices have an impact on levels of performance. Setting benchmarks for these performance indicators will provide guidelines for the establishment of developmental goals and extension advisory services toward an improved and efficient on-farm breeding system in smallholder farms.

## 2. Materials and Methods

### 2.1. Ethics Study Areas

The Ethics Committee (AEC) of the University of Pretoria (NAS339/2020) granted ethical approval for the use of external data. The current study is a sub-project of the High Value Beef Partnerships (HVBP) project funded by the Australian Centre for International Agricultural Research (ACIAR). The HVBP project (LS-2016-276) is a multi-provincial project that provides opportunities for SA smallholder farmers to participate in the free-range beef cattle market targeting middle-higher income consumers. One of the prerequisites for the success of the HVBP project is the improvement of on-farm breeding systems in SA smallholder farms. Data of the current study provides baseline herd reproductive performance levels required for setting improvement goals as a starting point in building a cost-effective on-farm breeding system. Reproduction records for the current study were collected from five of the nine SA provinces (Eastern Cape, Free State, Limpopo, Mpumalanga and North West province). The provinces represented the central and eastern regions of the country.

The majority of participating herds in the central regions (Free State, Limpopo, Mpumalanga and North West province) are found in two intermixed agriculturally productive rangeland biomes, the Savanna and the Grassland [30]. The region occupies 487,535 km^2^ of land with average temperatures between 28 °C in summer and 23 °C in winter. The annual rainfall range between 632 to 1600 mm [20,31,32,33,34,35]. Herds in the eastern region (Eastern Cape province) were sampled from the Albany thicket, Nama-Karoo, Stromberg plateau grassland, Grassland and Savanna biome. Grassland and Savanna biome contributed to majority of the sampled herds in this region. The eastern region covers 129,825 km^2^ of land with 24 °C maximum temperature in summer and 19 °C minimum temperature in winter. The province receives annual rainfall between 400 to 600 mm [36]. A map of SA showing the provinces where data were collected is presented in Figure 1.

### 2.2. Sampling Procedure and Data Collection

A multi-stage sampling method was implemented for the selection of provinces, herds and breeding cows within the herds. The study provinces were selected at a national level from provinces contracted within the HVBP project. Participating beef cattle herds within study provinces were purposefully selected based on the availability of handling facilities where reproductive measures such as pregnancy diagnosis were possible, while breeding cows were selected with the requirement that they had previously given birth to a calf. Cow indicators for reproductive performance (i.e., PR, FC, DO and CI) were collected in 2018 and 2019. The PR was obtained through pregnancy diagnosis using a portable ultrasound scanner [monitor (Ibex pro, EI medical imaging, USA; transducer (5 MHz/12 cm depth)]. It was defined as the percentage of cows found pregnant from all the cows checked for pregnancy in participating herds during pregnancy diagnosis. Pregnancy diagnosis was performed after every five months for each cow for the duration of the project and gestation length for each pregnant cow was measured in months. Cows were defined as having experienced FC when they were diagnosed as pregnant to the first pregnancy diagnosis but open and not lactating at the final pregnancy diagnosis. The FC for this study was defined as the percentage of both abortion and calf mortality in a herd. That is the period from prior birth to up to the first 28 days of life. Calf mortality in the current study was recorded from birth to 21 days of life. However, peri-natal mortalities may occur from birth to up to 28 days of life, these are therefore the most vulnerable time for the calf survival in an extensive production system [37,38]. Gestation length and age of the last calf for each participant cow was used to estimate DO and CI. Indicator DO was defined as the number of days between calving and conception and CI was defined as the number of days between two consecutive calving events. The estimate for CI was calculated by adding the gestation length (remaining months to calving) with the age of the last calf in months and DO was estimated from subtracting gestation length to the age of the last calf. That is the differences between the gestation intervals from the birth month of the age of the last calf to the current gestation during pregnancy diagnosis. As a result of challenges on accurate recording of performance data by farmers, the indicators FC, DO and CI were estimates and modified into categories (Table 1). The variables CI and DO were divided into four groups (acceptable, concern, extended, and overly extended) to better understand the heterogeneity within smallholder farms and establish the range in which the majority of farms fell within. Additional data collected on each cow included: breed, age and parity. Breeds were recorded as ‘‘type’’ according to the strongest resemblance of a specific breed type (Appendix A). Cows were raised on natural pasture with no supplementation. The above measurements were collected from 40 herds, distributed as follows: 16 herds in 2018 ((Limpopo (4), Mpumalanga (9) and North West (3)) and 24 herds in 2019 ((Eastern Cape (12), Free State (2), Limpopo (2), Mpumalanga (6) and North West (2)). Herds were visited twice a year, in autumn (March-May) for pregnancy diagnosis and again in spring (September–November) to monitor confirmed pregnancies, record pregnancy losses and identify new pregnancies. In addition, the second on-farm visit in the second year (2019) included questionnaire-guided interviews with each farmer to collect information on herd management. Farmer demographics and farm information (e.g., gender, education, off-farm income, farm engagement (part-time or full time), type of farming, herd size), as well as reproduction management data (e.g., knowledge of body condition score (BCS) prior to breeding, culling old and non-productive cows, type of breeding season, records keeping and bull to cow ratio) were recorded (Table 2). Breeding seasons ranged from continuous to a defined breeding season according to the farmers’ herd management preferences. The following breeding seasons were identified and recorded: January–March, March–June, August–October, September–December, November–February and December–March depending on the farmers’ choice.

### 2.3. Data Preparation and Editing

The validity and quality control of data in this study were guided by the overall HVBP project specifications including (1) the ability of farmers to finish their cattle on natural pastures for three years to meet free-range market specifications, and (2) herd health in line with the department of agriculture and the Animal Diseases Act 35 of 1984. The Act state that herds that test positive for venereal diseases such as contagious abortion (CA+), trichomoniasis and campylobacter must be referred to the state veterinarian for further evaluation. Given this, herds with venereal diseases were excluded from the study post first collection until they were cleared by the state, which greatly affected the number of repeated measurements. The final specification was the market price of the animal at 420 kg live weight at 3 years of age presented to the farmers. Some farmers were in agreement with the market price and others were not. This resulted in withdrawals of some farmers from the project.

The above specifications influenced the amount of data collected for this study as herds withdrew voluntarily or owing to herd health challenges, making them unavailable for data collection follow-ups, as shown in Figure 2 below. As a result of the above explained challenges, 5 of 16 herds collected in 2018 were repeated in 2019. Data were pooled across five provinces to report reproductive performance across a broader geographic spectrum in order to represent national reproductive performance in smallholder farms. Furthermore, the study provides an insight into reproductive performance at a provincial level with selective provinces representing the central and the eastern regions. Provincial representation was based on provinces with six or more herds where data were successfully collected twice a year (Eastern Cape and Mpumalanga provinces). At a national level, PR proceeded with all 3694 records collected from 40 herds. Indicator DO, CI and FC were assessed on 1401 records from 24 repeated herds (Autumn and Spring collection). The provincial level continued with 1003 records from Mpumalanga (central region) and Eastern Cape (eastern region) provinces. The flow of data is represented in Figure 2.

### 2.4. Statistical Analysis

Data were analyzed using Statistical Analysis System (SAS) 9.4. Frequency tables were used for summary statistics to show average performance levels. Chi-square test was performed to test for equal proportions.

A multilevel logistic regression model with random effects was applied using GLIMMIX procedure to assess measures of association between management factors and performance indicators (PR, FC, CI and DO). The model included provinces as random effects and management factors were fitted as fixed effects. Farms were considered as the experimental unit. An empty unconditional model without any predictors served as the starting point for the modeling procedure. This model provided a general estimation of the reproductive performance (PR, FC, CI, and DO) for farms at a typical province and information regarding the performance variation between provinces. Afterward, the model-building process continued to include herd management variables as fixed effects while controlling for provinces to estimate factors associated with performance measures at a national level. The regression model computed a cumulative ordinal regression procedure for the indicators CI and DO and a binary logistic regression procedure for the indicators PR and FC to estimate management factors associated with performance indicators. The binary model was described as follows:$$In\left(\frac{P\left(Y_{ij=1}\right)}{\left(Y_{ij=0}\right)}\right)=a_{i}+\beta_{xij}+u_{ij}\;$$

*Y_ij_* is the binary indicator of the *i*th farm in the *j*th province, with *Y_ij_* = 1 representing the probability of success (pregnancy/loss) and *Y_ij_* = 0 otherwise. Additionally, $a_{i}$ is the intercept and β is the regression coefficient of the *xij* covariates. Furthermore, $u_{ij}$ is the random effect representing the effect of the *j*th province.

The cumulative logit procedure simultaneously estimates multiple equations for the comparison of the cumulative odds of high versus low CI and DO categories. For this study, the predictor variable CI and DO have four categories as follows:$$j=\left\{\begin{array}{c} Accepted\\ Concern\\ Extended\;\\ Overly\;extended \end{array}\right.$$
where the overly extended category represents high outcome category and accepted category represent low outcome category.

Therefore, the logits regression model used for CI and DO was defined as:$$\left(\frac{P\left(Y\geq j\right)}{1-P\left(<j\right)}\right)=a_{j+\beta_{x}+u_{j,\;\;}}\;\left(j=\left(1,2\ldots \ldots j-1\right)\right)$$
where *p* (*Y* ≥ *j*) is the odds of the event of the category *j* of a given predictor variable (CI and DO); *α_j_* is the intercept parameter and *β* is the vector of regression coefficients corresponding to *x* covariates and $u_{ij}$ is the random effect representing the effect of the *j*th province. The model specifies that the intercept parameter differs across all *j* categories; however, the *x* covariates remain constant. The odds of the highest *j* level category (overly extended) was used to compare with the lower level category (accepted). Variables included in the models are presented in Table 2.

**Table 2 animals-12-03003-t002:** Variables included in the regression model.

Variable	Description	
Gender	1 = male, 2 = female	
Farm engagement	1 = part-time, 2 = full time	
Education	1 = primary, 2 = high school, 3 = tertiary, 4 = no school	
Off-farm income	l = Employment, 2 = Social grant, 3 = pension and business	
Herd size (no = cattle)	1 = small (1–50), 2 = medium (50–100), 3 = large (100–200), 4 = extra-large (over 200)	
Type of farming	1 = mixed = livestock and crops, 2 = livestock = cattle, goats, sheep	
Bull to cow ratio	1 = ideal = (1:30), 2 = under = (1:15) and 3 = over = (1:70)	
Culling old and non-productive cow	1 = yes, 2 = no	
Body condition scoring prior breeding	1 = yes, 2 = no	
Keeping calving records	1 = yes, 2 = no	

#### Determining Targeted Achievable Levels of Performance

To benchmark useful targets for beef cattle performance indicators in smallholder farms, the 25th, 50th (median), and 75th percentiles were chosen as summary statistics for all performance indicators. The preferred 25% of the herd for each performance indicator was used to determine the target levels for that indicator. This was a value higher for the first quartile (25%) or third quartile (75%). For this study, the 25th (lower) percentile was the target achievable level of indicators FC, DO and CI, while the 75th percentile (higher) value was the target achievable level for PR [39].

## 3. Results

The summary of reproductive performance records in smallholder herds at national level is presented in Table 3. Overall, majority of smallholder herds recorded 50% PR with 12% FC (abortion and calf mortality) and high CI (62%) and DO (39%) in the overly extended category (>608 and >304 days) (Figure 3).

Table 4 present a summary of reproductive performance records in smallholder herds at provincial level. Overall, PR yielded 61% with FC of 10% and majority of the herds recorded overly extended CI (55%) and DO (46%) days (Figure 3).

Summary of incidence of FC at provincial and national level is presented in Table 5. The chi-square test of equal proportions showed that, the incidence of FC was higher in cows that calved and lost the calf compared to aborted cows (*p* < 0.01). The majority of calves died during the 1–7 days period (national (5%) and (4%) provincial level) compared to during 8–14 days (national (3%); provincial (3%) and 15–21 days (national (1%); provincial (1%) (Table 5).

Figure 4 presents the interaction between breeding season and performance indicators (PR and FC) in smallholder farmers. Majority of incidences of FC and non-pregnant cows in the herds occurred during continuous breeding season as opposed to defined breeding season.

Table 6 represent the unconditional logistic regression model to test for the likelihood and variation of performance indicators between provinces at provincial level. The model revealed that at the provincial level, the probabilities of PR and FC are 0.62 and 0.09, respectively. Moreover, the cumulative likelihood of being in the overly extended CI and DO versus the accepted level were 0.69 and 0.89, respectively. The model shows no significant difference among provinces (*p* > 0.05), indicating that the likelihood of the performance indicators is constant across Eastern Cape and Mpumalanga province. Similar to the provincial level, there were no significant differences between the provinces (Eastern Cape, Free State, Limpopo, Mpumalanga and North West) on performance indicators at the national level (*p* > 0.05). The model revealed that the probabilities of PR and FC are 0.48 and 0.13, respectively. Moreover, the cumulative likelihood of being in the overly extended CI and DO versus the accepted level were 0.92 and 0.83, respectively, as shown in Appendix A.

The target level of performance for PR was (54%) at the 75th percentile and FC recorded (1.4%) at the 25th percentile. At the 25th percentile, DO and CI target levels yielded 152 and 425 days, respectively (Table 7).

Table 8 represents tests of association between herd indicators and household characteristics. There was no association (*p* > 0.05) between gender, farm engagement and off-farm income with PR within herds. Performance indicator DO was significantly different between different off-farm income (*p* < 0.05). An association was observed between CI and education level, off-farm income, herd size (*p* < 0.01), and gender (*p* < 0.05). Furthermore, FC was not different between different gender however, different (*p* < 0.01) between off-farm income and herd size.

The logistic regression model analysis for the relationship between management factors and performance indicators is shown in Table 9. There was an association between PR and culling old cows (*p* < 0.0022), and BCS prior breeding (*p* < 0.033). There was an increase in the odds (OR = 3.078) of FC for farmers who do not practice BCS prior to breeding. Farms that do not cull old cows and do not practice BCS prior breeding with a low bull to cow ratio had an increase in the odds (OR = 2.263; 1.306 and 2.332) of overly extended CI. Similarly, ex-tended DO was observed on farms that do not practice culling non-productive cows [OR = 1.880] and where calving records are not kept (OR = 2.274).

## 4. Discussion

The objective of this study was to assess reproductive performance as defined by PR, FC, CI and DO in SA smallholder farms to benchmark reproductive performance. The study presented reproductive performance norms and benchmarks for reproductive performance of beef cattle managed on natural pastures at an extensive system in smallholder farms of SA. The study also provided insight on associations of farmers’ management practices within the recorded performance indicators. Smallholder farmers need these benchmarks to identify current management weaknesses on herd reproductive performance and to provide a structured approach in addressing areas requiring improvement. In the current research, herd management influenced benchmarks of performance indicators. Reproductive performance in the study was categorized by low PR, high FC, extended DO and CI.

The overall annual PR reported at both national and provincial level was comparable with those reported in Bangladesh, Brazil and SA [21,40,41]. This level of performance is lower than the >75% recommended achievable performance of PR for beef cattle at extensive systems in tropical regions such as Australia [39,42,43]. The causes of variation in performance may be explained by consequences of chosen management practices such as uncontrolled breeding season by majority of the smallholder farmers in this study. It is to note that continuous breeding season in the current study reported more non-pregnant than pregnant cows and high percentage of FC. This highlights management flaws and may reflect on the reported limited advisory and extension services on farm management to smallholder farmers [44].

The current study reported FC losses that are consistent with the reports from past decade (12.83%) in smallholder beef cattle farms of SA [21]. This amplifies no improvement within the past decade and a half. South Africa is reporting annually higher losses than countries such as Brazil 4.1% and Portugal 5.7% [45,46]. Records in these countries may be influenced by openness to adoption of developmental programs such as the Welfare Assessment Protocol applied in New Zealand and Namibia. Application of this protocol assists in combating reproduction failures and the aforementioned countries are currently achieving <2.5% losses [27,47]. Similar to the current study, Australia reported majority of the losses to have occurred in the first week of calving in an extensive production system [38]. This area signifies the need for improvement to reduce calf mortality and improve weaning rates according to the recommended 2% pre-weaning mortality rate for beef cattle by the department of agriculture in SA [15].

Calving interval of 365 days for extensive beef cattle breeds in Southern Africa has been reported as impractical due to environmental stressors, therefore, a more reasonable range in this region may fall within 398 to 477 days [48]. This is in agreement with the targeted level derived for smallholder farms in this study. However, 75% of the herds in the current study obtained extended CI and DO (608 and 334 days), respectively, as achievable levels. This indicates that re-conception is potentially one of the major areas that require significant management interventions. The extended CI and DO highlight that farmers are either not aware of the cost to infertility or may not have the necessary skills and knowledge to manage it. Shortening these periods through better management can be beneficial on production and subsequently increase herd profit [39]. The study further revealed that farmers’ decisions of not culling old and non-productive cows, and not recording animal performances in herds needs to be revised as it consequently puts smallholder farmers at the 75th percentile for extended DO and CI periods. Amongst current management practices in smallholder farms, lack of knowledge of BCS prior to breeding by farmers in the current study was associated with increased FC and extended CI levels. The report of [49], indicated that for each BCS lost, postpartum anestrus is extended by 43 days and cows were further subjected to pregnancy losses [50]. Moreover, the study of [50] indicated that cows under 2–3 BCS of a five point scale was associated with the highest (14.91%) pregnancy losses in dairy cattle. That is, postpartum nutritional deficiency in cattle may impede uterine involution and expose cows to metabolic and infectious diseases which may result in pregnancy failure [51]. Therefore, a shift in management and receptivity to development interventions should be prioritized. A report on Indonesian beef cattle by [52], suggested that cost-effective interventions such as a defined breeding calendar, suckling restriction period, and pre and post-calving nutrition should be implemented for fertility improvement. A breeding calendar that is in concurrence with the rainy season is of most importance and can assist balance peak nutritional demands with the provision of enough grazing pasture preferably at late pregnancy and early lactation to promote re-conception [53,54]. Moreover, training of BCS and the importance of supplementation to maintain BCS primarily at the beginning of the breeding season to support pregnancy requirements and post-calving for support of estrus is encouraged [55]. These interventions may not only assist beef cattle smallholders in SA but other tropical countries such as Somalia, Vietnam, and Indonesia reporting similar results [56,57,58].

Record keeping is critical for analyzing areas of concern affecting farm growth. The present study has found that overly extended DO results from no record keeping. This expands the need for more awareness efforts emphasizing the importance of excellent record keeping towards the establishment of farm improvement [59]. Recording systems are gradually introduced in developing countries from paper to digital applications. The beef cattle farm management recording system (BCFM) in Thailand and the SA Long-term EU-Africa research and innovation Partnership on food and nutrition security and sustainable Agriculture (LEAP Agri) project is to gain popularity in smallholder farmers as a tool for record keeping [60,61]. These tools are to encourage farmers in collecting data and keeping up to date with farm productions in their pockets. Moreover, participation of farmers in programs such as the Agricultural Research Council (ARC) Kaonafatso ya Dikgomo (KyD) (Animal Recording and improvement Scheme) program in SA will not only provide recording knowledge but also assist farmers to practice good animal husbandry [62]. A proper recording will alert the farmers to reproduction failures such as non-productive cows which contribute to the overcrowding of reportedly strained rangeland of SA smallholder farms [62]. The results of the current study highlighted no association between gender and majority of reproductive performance as compared to those reported by [63]. This highlights that determination and drives to achieve performance are not gender dependent and that women are just as capable as men unlike in previous report by [9] where men outperformed women by over 50% in farm production. The current study further showed that larger herd sizes are associated with increased PR; however, were associated with higher FC and longer CI. The increase in FC and extended CI may indicate the lack of knowledge on herd management on production outputs and that large farms can have high marketable outputs however when the farm is managed well and with appropriate expertise [64].

Initiatives such as the Integrated Village Management System (IVMS) in Indonesia and the community-based breeding programs have improved reproductive management in village farms [65]. These programs promote good husbandry practices such as supplementary feeding of cows during late pregnancy and early lactation, and weaning calves at 6–8 months old for maintenance of BCS to promote re-conception. Through the IVMS program, calving rate in Indonesia has increased by 70% and 13.43 months of calving interval is observed [53]. Additionally, in Bali through supplementation feeding of breeding cows, smallholder farmers improved re-conception to up to 20% [66]. It is the adoption of such initiatives in SA that can assist in improvements of beef cattle reproduction. Lastly, recognition programs for excellent herd performance of smallholder farmers can implement a change in attitude on management behavior thereby creating a sense of belonging and reflecting the importance of smallholder farmers’ contribution to the beef cattle industry.

In SA smallholder farms, strategies for improving herd reproductive performance in an extensive farming system may include: understanding the significance of breeding season and modifying breeding season to match the quality of summer grazing. Additionally, supplementary feeding especially for high demanding animals such as pregnant and nursing cows is encouraged. This is for the maintenance of BCS and reducing the re-conception norm of two years and more in SA. For farm decision-making, farmers should invest in keeping thorough breeding records, as it is crucial in identifying challenges such as old and non-production cows, moreover through recording herd improvements can be identified. Extension and advisory officers may convey the outcome of this study and provide improved herd strategic management through open platforms such as farmers’ days, workshops and farmers study groups. These platforms may also encourage interactions with farmers and strengthen information chain between extension officers and farmers.

## 5. Conclusions

The study found that SA smallholder farmers at national and provincial level achieved performance levels for PR within the 50–60%, FC in the 10–12% and extended calving and days open within 608 and 334 days, respectively. The present study was also able to highlight key areas that require attention, firstly the period between calving to re-conception since majority of the herds achieved extended CI and DO. Secondly, the period between 1–7 days post calving due to more calf losses recorded in the first week of calving and finally the practice of continuous breeding season necessitates attention because of an increased number of non-pregnant cows obtained by continuous breeding season. Furthermore, the targeted performance benchmarks in the study highlighted that optimal reproduction in smallholder herds can be possible however with sound management structure in place. That is a management system that takes account of non-productive cows, defined breeding season, record keeping and awareness of herd nutrition status primarily prior to breeding. The defined areas of concern in the study provide an opportunity for the industry’ s extension and advisory services to know where to start in making management interventions towards improving reproductive performance benchmarks. It is recommended that further studies should take into account animal risk factors and environmental factors to refine the herd reproductive performance benchmarks and provide more insight into the reproductive performance of beef cattle in smallholder herds.

## Figures and Tables

**Figure 1 animals-12-03003-f001:**
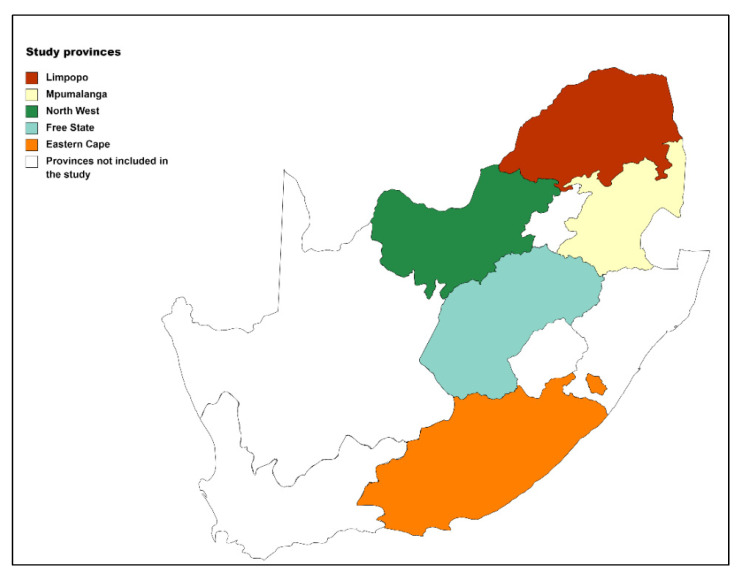
A map of SA showing the provinces where data were collected.

**Figure 2 animals-12-03003-f002:**
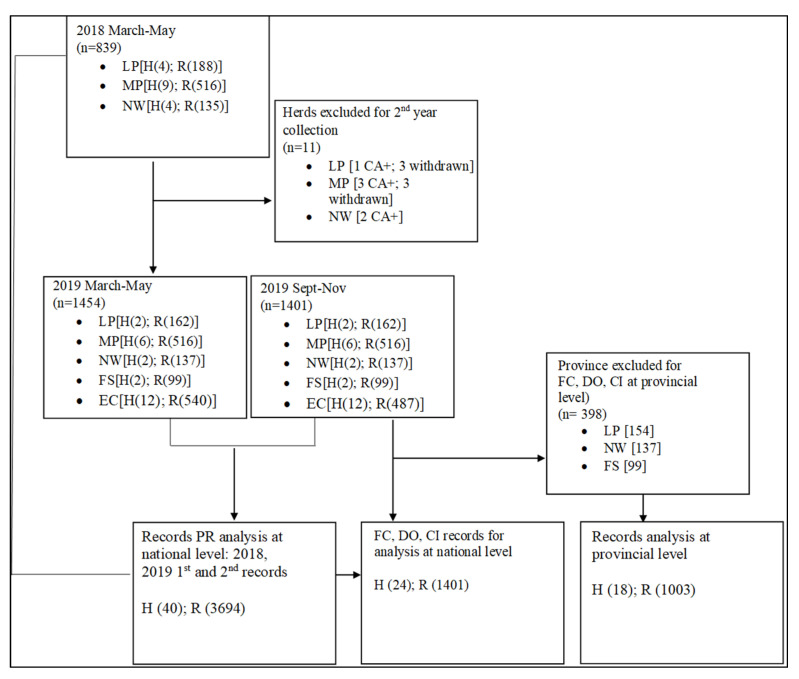
Demonstrate the flow of the data collected for the study in 2018 and 2019. Note: n = the number of records (R) from participating herds (H) in five provinces (EC = Eastern Cape; FS = Free State; LP = Limpopo; MP (Mpumalanga) and NW = North West) prior analysis of PR, FC, CI and DO.

**Figure 3 animals-12-03003-f003:**
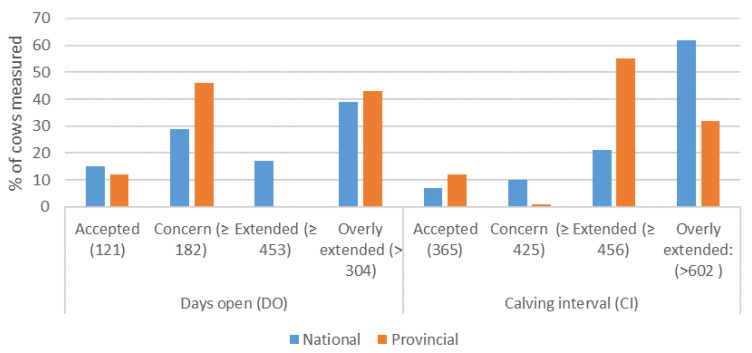
Measures of DO and CI levels in beef cattle farms at national and provincial level. DO and CI are recorded in days. The blue bar = DO and CI at national level and the orange bar = provincial level.

**Figure 4 animals-12-03003-f004:**
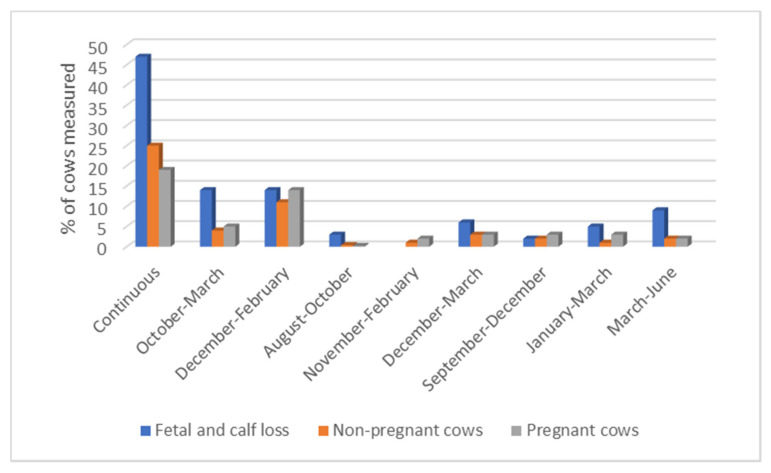
Occurrence of FC, non-pregnant and pregnant cows by breeding season.

**Table 1 animals-12-03003-t001:** Categories of reproductive performance indicators.

Indicators	Categories	Duration (Days)
PR	Pregnant	-
	Not pregnant	
FC	Aborted	-
	Calf mortality	1–7
		8–14
		15–21
DO	Accepted	121
	Concern	≥182
	Extended	≥243
	Overly extended	>304
CI	Accepted	365
	Concern	≥425
	Extended	≥456
	Overly extended	>608

**Table 3 animals-12-03003-t003:** Summary of reproductive performance of smallholder beef cows at national level.

Parameter	Herd	% of Parameter Measured
PR	40	50
CI	24	62
DO	24	39
FC	24	12

Note: % of parameter measured is the frequency % of the performance indicators (PR, CI, DO and FC).

**Table 4 animals-12-03003-t004:** Summary of reproductive performance of smallholder beef cows at provincial level.

Reproduction Parameters	Herds	% of Parameter Measured
PR	20	61
CI	20	50
DO	20	39
FC	20	10

Note: % of parameter measured is the frequency % of the indicators (PR, CI, DO and FC).

**Table 5 animals-12-03003-t005:** Occurrence of fetal and calf losses in beef cattle smallholder herds.

	No. Cows Pregnant	No. Cows Calved (%)	No. Cows with FC (%)	Period of FC (%)				*p*-Value
		Calving Records		Aborted	1–7 Days	8–14 Days	15–21 Days	
National	918	805 (88)	113 (12)	35 (4)	45 (5)	23 (3)	10 (1)	<0.0001 **
Provincial	691	620 (90)	71 (10)	15 (3)	30 (4)	17 (3)	9 (1)	<0.0001 **

Note: Statistically significant at level (** *p* < 0.01; * *p* < 0.05).

**Table 6 animals-12-03003-t006:** Summary of the likelihood and variation of reproductive performance of smallholder beef cattle herds at provincial level (Mpumalanga and Eastern Cape).

			95% CI				
Indicator	Estimate	Standard Error	Lower	Upper	*p*-Value	PP	Variation
PD					0.1856	0.62	0.02
Mpumalanga	0.2539	0.202	−0.1439	0.6518	0.2107		
Eastern Cape	−0.2555	0.2028	−0.6535	0.1425	0.2080		
FC					0.1869	0.09	0.13
Mpumalanga	0.6156	0.4821	−0.3310	1.5621	0.2021		
Eastern Cape	−0.5853	0.4806	−1.5289	0.3582	0.2236		
DO					0.2614	0.89	0.04
Mpumalanga	0.09073	0.1067	−0.1187	0.3002	0.3955		
Eastern Cape	−0.09099	0.1069	−0.3007	0.1187	0.3948		
CI					0.3324	0.69	0.02
Mpumalanga	0.2600	0.2036	−0.1395	0.6594	0.5931		
Eastern Cape	−0.2602	0.2036	−0.6596	0.1393	0.5930		

Note: Statistically significant at level (*p* < 0.05). SE = Standard Error, PP = predicted probabilities, CI = confidence interval.

**Table 7 animals-12-03003-t007:** Target level of reproductive performance in smallholder herds at 25th to 75th percentiles.

Parameter	No. Records	No. Herds	25th Percentile (Lower Quartiles)	50th Percentile (Median)	75th Percentile (Upper Quartiles)	Target Level
PR	3694	40	40	44	54	54
FC	918	24	1.4	1.9	2.4	1.4
DO	1344	24	152	212	516	152
CI	1344	24	425	516	608	425

Note: Target level is the level of performance based on either 25th or 75th quartile.

**Table 8 animals-12-03003-t008:** Summary of association between herd dynamics and the odds of performance in smallholder beef cattle.

Parameters	Gender	Education	Off-Farm Income	Herd Size	Farm Engagement	Type of Farming
PR	NS (0.7289)	<0.0001 **	NS (0.0581)	0.0092 **	NS (0.3886)	<0.0001 **
OR	0.964	3.044	1.061	1.115	1.116	1.838
FC	NS (0.0696)	NS (0.7491)	<0.0001 **	0.0003 **	NS (0.2469)	NS (0.1173)
OR	3.112	0.857	4.560	0.347	1.831	0.420
DO	NS (0.1595)	NS (0.1604)	0.0302	NS (0.1301)	NS (0.9747)	NS (0.5246)
OR	1.504	1.531	2.580	1.170	0.991	1.170
CI	0.0025	<0.0001 **	<0.0001 **	<.0001 **	NS (0.3317)	0.0216 *
OR	2.937	4.078	0.717	0.333	1.418	1.931

Note: OR = odds ratio, Significant at (** *p* < 0.01), (* *p* < 0.05) and NS = not significant at *p* > 0.05.

**Table 9 animals-12-03003-t009:** Summary of association between management factors and the odds of performance in smallholder beef cattle.

Management Variables	Indicators			95% CL		
		OR	SE	Lower	Upper	*p*-Value
	PD					
Culling old cows						0.002
YES vs. No		0.667	0.1323	0.515	0.865	
NO		Ref				
BCS prior breeding						0.033
YES vs. No		1.362	0.1452	1.025	1.811	
NO		Ref				
	FC					
BCS prior breeding						0.039
NO vs. YES		3.078	0.5442	0.05621	2.1922	
YES		Ref				
	CI					
Culling old cows						0.002
No vs. YES		2.263		1.341	3.819	
YES		Ref				
BSc prior breeding						<0.001
NO vs. YES		1.306	0.2421	0.191	0.493	
YES		Ref				
Bull to cow ratio						<0.001
Bull to cow ratio 3 vs. 2		2.332	0.1736	0.9500	1.6313	
Bull to cow ratio 1 vs. 2		0.275	0.7089	0.7093	1.6605	
	DO					
Culling non-productive cows						0.002
NO vs. YES		1.880		−0.1818	0.6191	
YES		Ref				
Calving records						0.005
NO vs. YES		2.274	0.2363	0.277	0.699	
Yes		Ref				

Note: Statistically significant at level (*p* < 0.01; *p* < 0.05). SE = Standard Error, OR = odds ratio, CI = confidence interval.

## Data Availability

Not applicable.

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
