# Peer review of "Assessing Reproductive Performance to Establish Benchmarks for Small-Holder Beef Cattle Herds in South Africa"

_animals, 2022, doi:10.3390/ani12213003_

Round 1
Reviewer 1 Report (New Reviewer)
The research aims at assessing reproductive performance as defined by pregnancy rate (PR), fetal and calf losses (FC), calving interval (CI) and days open (DO) on beef cattle farms to set benchmarks for herd reproductive performance, and assess whether management practices have an impact on levels of performance, it has very important guiding significance for smallholder farms. Data acquisition and analysis are the core of this study, and there are several places that need further explanation.(1) Supplementing information such as breed, age, parity and feeding conditions of cattle;(2)How to consider the influence of region, climate, parity, age and feeding mode on the results?(3)Whether specific suggestions can be given to farmers.
Author Response
Thank you for your valuable time and the comments towards improving our manuscript. Find the attached point by point responses.

Reviewer 2 Report (New Reviewer)
The manuscript reports on the study performed to examine reproductive performance as defined by pregnancy rate, fetal and calf losses, calving interval, and days open to benchmark smallholder herd reproduction. The study showed that some herd management practices and the lack of appropriate advisory and extension services are significantly associated with the recorded reproductive performance in smallholder farms.
The article is logic and well written. It appears to the reviewer that the justification for this work is generally well documented and scientifically receivable given. The experimental design along with the organization and interpretation of data is generally appropriate.
The only thing I do not understand is the presence of highlighted lines in the text and tables.
Suggested corrections:
Line 25: should be 1.4%
The two tables are named as "Table 7"
Tables 7-bis (8?) - Some numbers on the same line in italics, elsewhere not in italics, is this an editing error?
Line 458: explain ‘SA LEAP Agri”
Author Response
Thank you for your valuable time and comments provided. Please find the attached responses

This manuscript is a resubmission of an earlier submission. The following is a list of the peer review reports and author responses from that submission.
Round 1
Reviewer 1 Report
Assessing reproductive performance benchmarks for smallholder beef cattle herds in South Africa
The authors analyzed data collected from 40 smallholder beef cattle farmers in in South Africa to assess their reproductive performance and to establish future performance benchmarks. The authors established these benchmarks based on the 25th or 75th percentiles of the respective reproductive performance variables and evaluated reproduction data at the national and provincial level. The authors concluded that herd fertility was associated with certain management practices, such as poor culling practices, and that the decided benchmarks may be used to improve future reproductive performance. This data is very interesting and will be useful to outreach/ education personnel in South Africa. Nevertheless, changes need to be made to improve the readability and overall quality of this manuscript before it is suitable for publication.
General Comments:
• The title makes it seem as though the benchmarks themselves are being assessed instead of the reproductive performance to establish these benchmarks. I suggest revising the title to reflect this for clarity.
• The grammar and sentence structure of this manuscript make it difficult to read or clearly understand at times. Therefore, editing of English language and style is required throughout the manuscript. Some examples: plural vs. singular use of words (e.g. Line 180: “tests”), incorrect word usage (e.g. Line 62: “In South African smallholder farmers”; Line 73: “reproduction diseases” should be ‘reproductive’), short sentences could be combined (e.g. Lines 90 to 92).
• Certain abbreviations need revision. After establishing an abbreviation make sure to continue using it throughout the manuscript (e.g. PR is referred to as pregnancy rate multiple times after abbreviation was already established).
• If an abbreviation is only going to be used once or twice there is no need for an abbreviation (e.g. Quality control (QC) on lines 176 and 204; SDGs on line 58).
• There is no mention of any hypotheses throughout this manuscript.
• Inconsistencies with p-value reporting. E.g. (P<.0001) on line 46, (p < 0.05) on line 215, (p<0.01) on line 282, (P≤0.001) on line 307 etc. Make sure spacing, use of zero before the decimal point, and capitalization of ‘P’ are the same throughout the manuscript. Also take into account spacing around values and symbols (e.g. Line 64: ≤48%)
• Is this truly a multi-stage sampling approach?
• Suggest being consistent with number of decimal places throughout the manuscript. E.g. Line 41: 50% vs. Line 42: 12.31%
• Were any of the same herds used in both 2018 and 2019? Please make clearer.
• Please add standard deviations or standard errors for values (e.g. 62 ± 0.5 %)
• The results can be confusing at times. Clearer explanations are required.
• Discussion should be improved. Better connections to the results of this study should be
made. Associations need to be discussed.
Specific Comments:
Line 25 and 26: Brackets should be removed
Line 40 and 232: The “best” 25th quartile should be revised since best could be subjective.
Line 54: “important backbone” should be revised since both words imply importance.
Lines 114, 117,157, 158 etc.: Suggest naming provinces in alphabetical order.
Line 124: Eastern does not need to be capitalized
Line 126: Suggest: A map of the provinces from which data were collected are presented in
Figure 1.
Figure 1: Suggest changing the figure key since the “Other Provinces” are included under the
heading study provinces. Perhaps change the heading from “Study Provinces” to “Provinces” or
change “Other Provinces” to “Provinces not included in the study”/ remove “Other Provinces”
from the key.
Line 130: Suggest: A map of South Africa showing the provinces where data were collected.
Line 138: “bred a calf” is incorrect. Had previously given birth to a calf? i.e. They were not
heifers?
Line 139: Abbreviations are made here but these are not the first time these words are used
(e.g. Lines 81 and 96).
Line 141: Include details on ultrasound device such as make/model/company/ frequency of
probe etc. if possible. Could combine this sentence with the next.
Line 141: This definition needs to be made clearer. Were all cows in the herd preg checked?
Line 142: Suggest: Cows were defined as having experienced foetal loss when they were
diagnosed as pregnant to the first pregnancy diagnosis but open at the final pregnancy
diagnosis. When were the diagnoses conducted? Is it possible to know an average age of
pregnancy at the time of diagnosis?
Line 144: Foetal loss should not include calf loss after birth since it refers to a foetus. I suggest
creating 2 separate categories. i.e. foetal loss and calf loss since they are different and may be
influenced by different factors.
Lines 147 and 153: These sentences both speak about CI. Suggest moving them closer together
so flow is better.
Line 150: How were these categories chosen?
Line 163: Remove extra space before bracket.
Line 165: Body condition score (BCS)
Table 1: This table is a bit confusing. Is the “Aborted” Category in the Pregnancy Rate or Foetal
Loss Indicator? It is unclear. For the calf mortality category the duration of ≤ 21 could
technically include animals from the other durations too. Should this be 15 to 21? Suggest
adding spaces or lines to separate different indicators. The Categories in Days Open and Calving
Interval are too close together making it more difficult to follow. How were these durations
determined to be Accepted/Concern/Extended/Overly Extended? Should these be Acceptable/
Of Concern?
Line 184: What does 3) refer to? Is it connected with the 1) and 2) above? If so perhaps
separate 1) and 2) in different sentences. Otherwise it seems somewhat random.
Line 195: Twice a year
Line 196: What do you mean by pregnancy rate continued?
Figure 2: This figure is confusing to interpret at first and should be improved. Certain boxes
have extra space for no reason and boxes are not consistent.
Line 205: MP = Mpumalanga. Spacing around equal signs not consistent.
Line 212: Info in two brackets can be combined. E.g. (P < 0.05; Table 2). Other instances in the
manuscript.
Line 213: See above comment on calf mortality
Line 227: Remove extra space by X2
Table 2: These categories need a better description in the M&M section for clarity. What are
the bull to cow ratios considered to be ideal/under/over? More information needs to be
provided in the M&M section since these categories can be subjective.
Line 262: Again, I think that foetal loss is an incorrect description of what you measured. You
either need to change the name or consider separating data into two different variables.
Lines 263: This is confusing to read, please edit for clarity.
Table 3: This table needs to be edited. PR for 2018 should above 2019. There shouldn’t be an
entire column for Overall PR that stretches over all the variables. Instead insert another row for
this information. Where CI/DO/foetal loss not recorded in both 2018 and 2019? If splitting PR
into 2 years why not for other variables?
Figure 2: The x-axis categories need to be edited. What does Accepted/Concern mean? Please
include this information in the figure description. What do the blue and orange bars represent?
It is unclear.
Table 4: Use the same number of decimal places for Eastern Cape and Overall columns. Include
zero before decimal point of p-values.
Line 282-284: Missing %, combine brackets with Table 5
Table 5: Missing %
Figure 3: This figure should be bar graphs and not a line graph since x-axis categories are not
related.
Table 6: Consistent decimal places. Target level is confusing. In the note it says that it is the
recommended level of performance but it is only going to be recommended in the future and
not at present. Therefore, it should be changed to something along the lines of target level of
performance selected based on either 25th or 75th quartile. Or at least describe it in a better
way that they have been selected as recommended baselines for the future.
Table 7: This table provides p-values but there are no indications of what type or size of the
association. You need to provide more information for the reader. Are the associations positive
or negative? Are they large or small?
Line 317: Table 8
Table 8: There is no heading for column 2.
Line 329: Please correct p-values
Line 333: “We” should not be used
Line 345: What does “>75% proposed guidelines” mean? Please be more clear
Line 353: What losses? Please specify. Also, there is mention of “reports” but only one
reference is provided. Are there more that could be added?
Line 356: reference 45 does not mention Portugal. Please provide an additional reference.
Line 356-358: There is mention of improvements in countries (Brazil and Portugal) but no
evidence of an improvement.
Line 360: Move reference to end of sentence
Line 374: Additional discussion could be added on importance of short CI and fewer DO.
Line 376: How did the lack of knowledge contribute? Please make the connection
Line 385 and 386: Remove extra spaces
Line 414: There is discussion about supplementation but there was no supplementation in the
current study. Please link the discussion of supplementation better to the current study.
Line 422: Within the 50% mark of what? Why is there a range for foetal loss but not the other
variables?
Lines 426-428: This is confusing to read. Please make clearer.
Author Response
Thank you for your valuable time and effort you took to review our manuscript. Attached are our responses to the comments and questions

Reviewer 2 Report
I have reviewed animals-1760899 titled "Assessing reproductive performance benchmarks for small-holder beef cattle herds in South Africa". The subject is within the area of the journal. However, there are many points need to be improved, and it is unsuitable for publication in its present form. I have many comments to the manuscript, but in this step my general comments and considerations are enclosed below. I will wait with specific comments until the authors have investigated my concerns.
l My biggest concern is about data analysis. Authors used Chi-squared test to evaluate the association between reproductive performance and each variable, but I recommend using multiple logistic regression analysis which make it possible to take into account the relationships among variables. In addition, current data analysis did not assess the interaction between variables. Furthermore, data used in this study were collected from multiple farms, and the effect of farms should be included in multiple logistic regression analysis as random effect.
l Regarding pregnancy rate and foetal loss, the observational unit was farm or cow? As mentioned in line 141, "the percentage of cows found pregnant within the herd" indicates the observational unit is "farm", but data for PR was 3694 records. It makes me confused. The definition of PR and foetal loss should be described more detail, how you classified cows as PR and foetal loss positive or negative.
l Regarding foetal loss, what is the difference between foetal loss and calf mortality? In general, foetal loss indicates embryonic death, indicating abortion, but I guess this variable would be calf that born alive and dead in early stage of life. Please clarify this point. Additionally, if this measurement is a calf record, you should add the information about calf records.
l Regarding DO and CI, these measurements are continuous variables, but no description about statistical analysis was found in the manuscript. As well as my comments above, this analysis should take into account multiple analysis, interaction and the effect of farm.
l Who answered farm information? Owner or employee? How did you treat gender, farm engagement, education and off-farm income if there are many employees?
l Herd size indicates number of cows or included the other type of cattle?
l Please define how to choose bull to cow ratio: ideal, under and over.
l The item of "culling cow" and "BCS prior breeding" are cow-based question and cannot select Yes or No as farm-level. How did you treat it?
Author Response
Thank you for your valuable time and efforts you took in reviewing our manuscript. Please find the attached responses to the comments and questions

Round 2
Reviewer 2 Report
Based on your responses to my previous comments, I have additional comments to the manuscript. There are still many points need to be improved, and I will wait with specific comments until the authors have investigated my concerns.
l I understand the observational unit was farm. In this case, you should delete the number of individual records in the all tables because it make readers confused.
l Additionally, I am in doubt about the soundness of the statistical methods used in Table 4 and 5. You stated the observational unit was farm, then there seems to be no statistical significance because the sample number was 40 and there was limited difference. You should evaluate it again. If the observational unit was individual records of cow, you should use mixed effect regression analysis.
l As well as my comments above, what is the observational unit in Table 7? If the farm, the number of observations was 40? In addition, chi-square cannot estimate the association. What kind of statistical method used?
l In Table 8, the number of observations exceed 40 and what is the observational unit?
I cannot recommend publication until these points become clear.
Author Response
We thank the reviewer for their time and efforts in reviewing our manuscript, we hope the responses we provided will provide clarity where is due.

Round 3
Reviewer 2 Report
Based on their responses to my previous comments, the experimental design in this study was not valid. Therefore, I recommend a manuscript for rejection.